# Impact of Guidelines Regarding Dihydropyrimidine Dehydrogenase (DPD) Deficiency Screening Using Uracil-Based Phenotyping on the Reduction of Severe Side Effect of 5-Fluorouracil-Based Chemotherapy: A Propension Score Analysis

**DOI:** 10.3390/pharmaceutics14102119

**Published:** 2022-10-06

**Authors:** Nicolas Laures, Céline Konecki, Mathias Brugel, Anne-Lise Giffard, Naceur Abdelli, Damien Botsen, Claire Carlier, Claire Gozalo, Catherine Feliu, Florian Slimano, Zoubir Djerada, Olivier Bouché

**Affiliations:** 1Department of Gastroenterology and Digestive Oncology, CHU Reims, University of Reims Champagne-Ardenne (URCA), 51100 Reims, France; 2Department of Medical Pharmacology, University of Reims Champagne-Ardenne (URCA), HERVI EA3801, 51097 Reims, France; 3Department of Pharmacology and Toxicology, CHU Reims, 51100 Reims, France; 4Department of Hepato-Gastroenterology and Digestive Oncology, Centre Hospitalier Auban-Moët, 51200 Epernay, France; 5Department of Hepato-Gastroenterology and Digestive Oncology, Centre Hospitalier de Chalons en Champagne, 51000 Chalons en Champagne, France; 6Department of Pharmacy, CHU Reims, University of Reims Champagne-Ardenne (URCA), 51100 Reims, France

**Keywords:** fluorouracil, dihydropyrimidine dehydrogenase deficiency, adverse effects, uracil

## Abstract

Dihydropyrimidine dehydrogenase (DPD) deficiency is associated with severe fluoropyrimidines-induced toxicity. As of September 2018, French recommendations call for screening for DPD deficiency by plasma uracil quantification prior to all fluoropyrimidine-based chemotherapy. A dose reduction of fluoropyrimidine is recommended when uracil concentration is equal to or greater than 16 ng/mL. This matched retrospective study assessed the impact of DPD screening on the reduction of severe side effects and on the management of DPD-deficient patients. Using a propensity score, we balanced the factors influencing 5-Fluorouracil (5-FU) toxicity. Then, the severity scores (G3 and G4 severity as well as their frequency) of patients who did not benefit from DPD screening were compared with those of patients who benefited from DPD screening for each treatment cycle (from 1 to 4). Among 349 screened patients, 198 treated patients were included. Among them, 31 (15.7%) had DPD deficiency (median uracilemia 19.8 ng/mL (range: 16.1–172.3)). The median toxicity severity score was higher in the unscreened group for each treatment cycle (0 vs. 1, *p* < 0.001 at each cycle from 1 to 4) as well as the cumulative score during all courses of treatment (*p* = 0.028). DPD-deficient patients received a significantly lower dose of 5-FU (*p* < 0.001). This study suggests that pretherapeutic plasmatic uracil assessment, along with 5-FU dosage adjustment, may be beneficial in reducing 5-FU toxicity in real-life patients.

## 1. Introduction

Fluoropyrimidines, mainly represented by 5-fluorouracil (5-FU) and its oral prodrug capecitabine, are antineoplastic agents with a significant role in the treatment of cancer [1,2,3,4,5]. Fluoropyrimidines are the backbone of the oncological treatment in numerous localisations (digestive, breast, and head and neck cancers), in all settings (neoadjuvant, adjuvant, or palliative). They are frequently used in association with other antineoplastic agents or radiotherapy. As analogues of uracil, fluoropyrimidines block the thymidylate synthetase and disrupt DNA synthesis [1]. They are eliminated through the classical endogenous pyrimidines (uracil, thymidine, and cytosine) metabolisation pathway. 5-FU is metabolised by the enzyme dihydropyrimidine dehydrogenase (DPD) to dihydro-5-fluorouracil, which is itself eliminated by urine [1,6,7]. The DPD activity is encoded on chromosome 22 by the *DPYD* gene and constitutes the keystone of fluoropyrimidine elimination. It is responsible for nearly 85% of their catabolism [8]. Therefore, DPD inhibition or deficiency leads to 5-FU accumulation. Deficiencies can be partial or complete, with a prevalence among Caucasians estimated between 3 and 8% and 0.01 and 0.5%, respectively [9,10,11,12,13,14].

According to the National Cancer Institute (NCI) Common Terminology Criteria for Adverse Event (CTCAE), grade ≥ 3 adverse events occur predominantly in rapidly renewing tissues, explaining the frequency of the digestive and haematological symptoms [15,16,17,18]. Fluoropyrimidine-induced severe toxicities have an estimated incidence between 10 and 30% [2,15,19,20]. By contrast, lethal toxicity is rare (incidence less than 1%) and occurs after multiple organ failure and aplasia [15,19].

Several studies have established a relation between the iatrogeny of fluoropyrimidines and the enzymatic deficiency of DPD, which can be observed in 20% to 60% of cases, depending on the study [8,9,11,20]. Serious adverse events are usually reported immediately after the administration of the first 5-FU dose [9]. Deficiency screening can be performed by genotyping (search for variants *DPYD*2A*, rs3918290; *DPYD*13*, rs55886062; c.2846A > T, rs67376798; c.1236G > A-HapB3; rs56038477) [16,20], or by phenotyping (i.e., measuring the enzyme functional activity) [15,17,18]. Enzyme activity can be estimated either by the dihydrouracil/uracil ratio (UH2/U) or simply by the measurement of the plasmatic uracil level [15,21] (recommended by the French Haute Autorité de Santé (HAS)). A recent study compared DPD deficiency screening by these three methods (genotyping, and phenotyping with only uracil concentrations or with the UH2/U ratio) and concluded that the mutation status of *DPYD* had a very low positive predictive value in identifying individuals with DPD deficiency [22].

Since September 2018, the French guidelines have recommended a DPD deficiency screening by plasmatic uracil quantification before any fluoropyrimidine-based chemotherapy prescription [23]. Similarly, in Belgium, phenotyping based on plasma uracil levels is also recommended. In April 2020, the European Medicines Agency (EMA) recommended genotyping and phenotyping based on plasma uracil levels to screen DPD deficiency [24]. A plasmatic uracil level over 150 ng/mL reflects a complete DPD deficiency leading to a contraindication for fluoropyrimidine due to a high risk of severe toxicity. When the level ranges from 16 ng/mL to 150 ng/mL, the deficiency is considered partial, associated with an increased risk of fluoropyrimidine toxicity. In this case, the initial fluoropyrimidine dose must be adapted and thereafter readjusted in the following courses of treatment, according to tolerance.

These new guidelines are easy to apply for patients with complete DPD deficiency. However, for patients with partial DPD deficiency these guidelines are less restrictive and may lead to different treatment adaptations.

The first objective of this study was to investigate the impact of the French guidelines concerning DPD deficiency screening using uracil-based phenotyping on the reduction of severe side effects of fluoropyrimidine-based chemotherapies. The secondary objective was to describe how these new guidelines are applied in daily practice for deficient patients.

## 2. Materials and Methods

### 2.1. Study Design and Patients

This French retrospective study was conducted in one tertiary oncology centre (Centre Hospitalo-Universitaire de Reims) and two secondary centres (Centre Hospitalier Auban-Moët d’Epernay and Centre Hospitalier de Châlons en Champagne). All subsequent patients over 18 years of age treated with 5-FU and who had an available pre-therapeutic uracil concentration measurement were included in the study. Data were collected between 4 October 2018 and 24 December 2019.

Patients were compared to a non-screened group, in which we included 100 patients randomly drawn from fluoropyrimidine-treated patients in 2017, prior to the HAS guidelines on DPD deficiency screening.

### 2.2. Demographic and Oncologic Data 

Baseline characteristics at first treatment, such as age, gender, anthropometric indicators (weight, height, body mass index (BMI)), estimated glomerular filtration rate (using the Chronic Kidney Disease EPIdemiology (CKD-EPI) algorithm), performance status and tumour location were collected and analysed from the electronic health medical records (eHMR).

### 2.3. Anticancer Treatment

Treatment characteristics were extracted from chemotherapy records (Chimio^®^ (V5.7 edited by Computer Engineering)) and eHMR. Data such as the type of chemotherapy (fluoropyrimidine alone or combined in a bi- or tri-chemotherapy regimen), its combinations with biotherapy or radiotherapy, and the setting of the prescription (neoadjuvant, adjuvant or palliative indication) were collected.

Different chemotherapy protocols were used:-LV5FU2 alone or included in a protocol with oxaliplatin (FOLFOX), irinotecan (FOLFIRI), or docetaxel (TFOX): leucovorin 400 mg/m^2^, followed by intravenous 400 mg/m^2^ 5-FU bolus and then continuous 5-FU infusion at the dose of 2400 mg/m^2^ over 46 h every 14 days; in some simplified FOLFIRINOX protocols, no bolus was administered.-LV5FU2-dacarbazine (DBZ) protocol: leucovorin 400 mg/m^2^, followed by intravenous 400 mg/m^2^ 5-FU bolus on days 1 and 2 and then continuous 5-FU infusion at the dose of 1200 mg/m^2^ over 46 h.

Dose modifications were also collected. The French recommendations advocate 5-FU doses tailoring according to the extent of the detected DPD impairment and adjusted based on age, general condition, and other clinical/paraclinical covariates if required. In our study, patients were adjusted as follows: full dose when plasma uracil < 16 µg/mL; 25% dose reduction when 16 µg/mL < plasma uracil < 50 µg/mL; 50% reduction when 50 µg/mL < plasma uracil < 100 µg/mL; 75% reduction when 100 µg/mL < plasma uracil < 150 µg/mL. The 5-FU dose intensity (DI) was calculated for patients with a determined phenotype. DI was obtained by dividing the cumulative dose by the planned duration of treatment (4 weeks) [25]. 

### 2.4. Chemotherapy-Induced Toxicities 

Clinical and biological chemotherapy-induced toxic manifestations were noted at each pre-therapeutic medical visit and after the assessment of the oncologist. The following events were reported and evaluated according to the NCI CTCAE criteria [26]: anaemia, neutropenia, thrombocytopenia, nausea, vomiting, mucositis, diarrhoea, alopecia, and hand-foot syndrome. We focused on grade 3 and 4 events.

Toxicities were managed by chemotherapy dose reduction, treatment postponement, or treatment discontinuation.

### 2.5. Time of Assessment

Assessment of 5-FU treated patients lasted four cycles, i.e., eight weeks of treatment. 

### 2.6. DPD Phenotyping

Uracil measurements were determined according to the latest French guidelines and performed in a single laboratory to limit inter-laboratory variability.

Preanalytical conditions for uracil concentration measurement were rigorous, implying the use of specific material (lithium heparinate sample tubes without separating gel), a limited time between sampling and centrifugation (<2 h), transport conditions respecting the cold chain and sample freezing (−80 °C) immediately after plasma separation.

Plasmatic uracil (U) and dihydrouracil (UH2) were quantified at the Department of Pharmacology and Toxicology, CHU Reims, using a sensitive ultra-performance liquid chromatography-tandem mass spectrometry (UPLC-MS/MS) method. Analytes were extracted by solid-phase extraction (SPE) using CX100-Interchim cartridges (Atoll, San Diego, CA, USA) according to the manufacturer’s instructions. The analytes were chromatographically separated on an Acquity UPLC HSS T3 column (Waters Corp; Milford, MA, USA) with an appropriate elution gradient and then analysed with a tandem mass spectrometer (Orbitrap QExactive, ThermoFisher Scientific, San Jose, CA, USA). U and UH2 were quantified in the positive ion mode. [^13^C,^15^N_2_]-Uracil (Alsachim, Illkirch Graffenstaden, France), the stable isotope for uracil, was used as an internal standard. The total chromatographic run time was 7 min. The validated concentration range for U and UH2 was from 3.13 to 200 µg/L. Inter-assay bias and precision for uracil ranged from −2.21 to 1.08% and from 2.18 to 7.20%, respectively. For UH2, inter-assay bias and precision ranged from −7.69 to −0.47% and from 5.65 to 8.43%, respectively. All assays were within the recommended limits (±15%) [27]. DPD deficiency was considered partial for U concentration values between 16 and 150 ng/mL, and total for values above 150 ng/mL.

### 2.7. Primary Endpoints

To investigate the impact of the French guidelines for DPD deficiency screening on the reduction of severe side effects of fluoropyrimidine-based chemotherapy, the factors influencing fluoropyrimidine toxicity were balanced using a propensity score. 

Then, a severe iatrogeny score, called the severity score, was determined to increase relevance and include frequency and severity. This score was the primary endpoint and was estimated for each group by the number of events, multiplied by 1 for grade 3 toxicity events, and by 2 for grade 4 toxicity events. 

### 2.8. Secondary Endpoints 

Deferral (postponement) of treatment (clinician’s choice not to adhere to the usual time between treatments, number of planned courses not completed, or prolonged expected time), and discontinuation of 5-FU (clinician’s choice to discontinue fluoropyrimidine treatment) were chosen as secondary endpoints.

Furthermore, the prevalence of DPD deficiency was estimated in phenotyped patients, and 5-FU treatment management in deficient patients (i.e., dose adjustment and relative DI) was described.

### 2.9. Statistical Analysis

Statistical analyses were performed with R 3.4.1. (the R Foundation for Statistical Computing, http://www.r-project.org, accessed on 15 May 2021). The Gaussian distribution of the data was explored using the Shapiro–Wilk test. Following these explorations, if the distribution was normal, the quantitative variables were compared with a Student’s *t*-test and, if not, with a nonparametric Mann–Whitney test. For the qualitative variables, the two samples were compared with a Chi2 test or an exact Fisher test, depending on group size. For continuous measurements, data are presented as mean ± standard deviation for normal distributions, or median with [min–max] for non-normal distributions. For qualitative parameters, data are presented as the number of cases *n* (percentage (%) of patients) 

We identified the two patient groups as DPD PG for the dihydropyrimidine dehydrogenase phenotyped (screened) group and DPD NPG for the dihydropyrimidine dehydrogenase non phenotyped (non-screened) group.

To control for confounding factors in the case of an unknown relationship between the DPD PG and DPD NPG groups and covariates, we used inverse probability weighting (IPTW). IPTW weights were estimated as the inverse of patients’ estimated probability of belonging to the DPD PG group. The ability to use all the individuals in the groups for the outcomes analysis is advantageous for propensity score weighting. All the individuals were included in the analysis, and the statistical power to detect the group effect was maintained. Propensity scores were calculated based on the probability of being in the two groups using generalised boosted models, a multivariate nonparametric regression technique [28,29,30]. This method is considered to be able to accurately estimate the propensity scores, even with large numbers of covariates. In the propensity score models, the group (DPD PG or DPD NPG) was the dependent variable, and all confounders were independent variables: age, gender, BMI, creatinine clearance, performance status, tumour type and stage (metastatic or not), chemotherapy, use of biotherapy and use of radiotherapy 

The covariate imbalance was assessed using standardised effect size (std.eff.sz) or standardised bias, defined as the treatment group mean minus the control group mean divided by the treatment group standard deviation, and by Kolmogorov-Smirnov (KS) statistics for each covariate as well as interactions. *p*-values for the KS statistics were derived from Monte Carlo simulations. All *p*-values were two-tailed, with statistical significance indicated by a value of *p* < 0.05.

### 2.10. Ethics

The study was conducted in accordance with the Helsinki Declaration. As the study was based on medical data systematically recorded for standard care at the Reims University Hospital and authorised by the French national commission for data privacy (Commission Nationale Informatique et Libertés, CNIL), the study did not require approval by an Ethics Committee according to French legislation on human research. Data were processed after anonymisation, and the database was created in accordance with the CNIL MR004 method (no. 2206749, 13 September 2018).

## 3. Results

### 3.1. Patients 

Among 349 patients screened by plasmatic U quantification, a total of 198 5-FU-treated patients were included during the study period. The mean age in this group was 66.4 ± 11.8 years. As a reference population, 100 patients were randomly drawn from patients treated by 5-FU in 2017, of which 6 were excluded. The mean age in the final reference group was 61 ± 12 years. For our analysis, we thus compared 198 patients who underwent DPD deficiency screening (DPD PG group) to 94 reference patients (DPD NPG group). The study flow chart is presented in Figure 1.

Clinical and treatment characteristics of the 292 included patients are described in Table 1. No difference in baseline characteristics was found for gender, anthropometric indicators (weight, height, and BMI), estimated glomerular filtration rate, performance status, and tumour location. However, the two groups’ ages, treatment types, and chemotherapy protocols were statistically different.

### 3.2. Comparison of Severe Toxicities

The median toxicity severity score was higher in the non-screened group than in the screened group during the first treatment cycle (1 [1.00;3.00] vs. 0 [0.00;3.00], *p* < 0.001), the second treatment cycle (1 [1.00;3.00] vs. 0 [0.00;2.00], *p* < 0.001), the third treatment cycle (1 [1.00;2.00] vs. 0 [0.00;2.00], *p* < 0.001), and the fourth treatment cycle (1 [1.00;1.00] vs. 0 [0.00;3.00], *p* < 0.001) (Figure 2 and Table 2) as was the cumulative score during all courses of treatment (*p* = 0.028) (Figure 3).

There was no significant difference between the two groups regarding the prevalence of at least one severe toxicity, which was estimated at 5.6% at treatment cycle 1, 4.2% at treatment cycle 2, 4.3% at treatment cycle 3, and 3.4% at treatment cycle 4 in the DPD PG group and at 8.5% at treatment cycle 1, 9.8% at treatment cycle 2, 9.8% at treatment cycle 3, and 4.4% at treatment cycle 4 in the DPD NPG group. Details about encountered acute severe toxicities are presented in Table 3. 

Concerning the frequency of treatment postponement, there was no significant difference between the two groups for the first and second treatment cycles (*p* = 0.313 and 0.606, respectively). Still, treatment postponement was significantly more frequent in the DPD PG group for the third course of treatment (*p* = 0.012) (Table 2).

Regarding discontinuation of treatment, there was no significant difference between the two groups after each treatment cycle (*p* = 0.723, 0.280, and 0.067 after the 1st, the 2nd, and the 3rd treatment cycle, respectively) (Table 2).

### 3.3. DPD Deficiency Prevalence

In the DPD PG group, 31 out of 198 (15.7%) patients were identified as DPD deficient. Among the DPD deficient patients (*n* = 31), median uracil concentration was 19 ng/mL (range: 16.1–52.2).

### 3.4. DPD Deficiency and Fluoropyrimidines Dose Adjustment

5-FU dose (bolus and infusion) was reduced for DPD deficient patients by 0 to 100% from the beginning of the treatment: the bolus dose was reduced in 81%, 62%, 48%, and 44% of patients (Table 4) and the infusion dose was reduced in 71%, 41%, 26%, and 20% of patients (Table 5) at the first, second, third, and fourth treatment courses, respectively.

There was no significant difference for the median 5-FU dose between the DPD PG and the DPD NPG groups: 2800 mg/m^2^ [400;4000] vs. 2800 mg/m^2^ [2000;2800], *p* = 0.060 for the first course, 2800 mg/m^2^ [0;4000] vs. 2800 mg/m^2^ [0;2800], *p* = 0.257 for the second course, 2800 mg/m^2^ [0;4400] vs. 2800 mg/m^2^ [0;2800], *p* = 0.352 at the third course, and 2800 mg/m^2^ [0;4000] vs. 2800 mg/m^2^ [0;2800], *p* = 0.146 at the fourth course (Table 2).

In the DPD phenotyped group, median dose-intensity was significantly lower in deficient patients (1056 ± 351 mg/m^2^/week) than in non-deficient patients (1233 ± 251 mg/m^2^/week) (*p* < 0.001).

## 4. Discussion

To our knowledge, this study is the first to evaluate the impact of DPD deficiency screening using uracil-based phenotyping on reducing the severe side effects of 5-FU-based chemotherapy.

First, this study showed, using real-life data, that the new French guidelines allowed a reduction in 5-FU serious toxic events during the first four courses of chemotherapy.

Usually, fluoropyrimidines induce severe toxicities in 10–30% of patients, and, according to the literature, 30–80% of these toxicities could be attributable to DPD deficiency [31]. We found the same type of severe toxicity as described in previous articles, such as myelosuppression, mucositis, diarrhoea, and nausea [32,33]. Our analysis identified a significant difference in adverse effects toxicity coupled with their frequency between patients with an identified DPD phenotype and patients with an unknown DPD phenotype. Severity scores were significantly reduced for each treatment administration and cumulatively.

Our study describes a lower prevalence of severe toxicity (5.6%, 4.2%, 4.3%, and 3.4% at treatment cycles 1, 2, 3, and 4, respectively) than what is usually described in the literature. Toxicity occurrences may have been underestimated by focusing on grade ≥ 3 side effects, which are more biologically or clinically relevant. As reported by several studies, the toxicities avoidable by phenotyping are those of the first and second cycles [18,32,34]. In our study, we identified a difference in toxicity score up to the fourth treatment cycle probably due to the accumulation of drug toxicities, the decrease of physiological capacity of cell regeneration related to the depletion of stem cells, haematopoietic progenitors, and the depletion of cellular detoxification systems against oxidative stress. No life-threatening events were observed in either group.

Treatment postponements and dropouts were significant, potentially leading to an increased risk of treatment failure. 

In our opinion, the main strength of this study is the large number of patients studied with homogeneous baseline characteristics such as gender, anthropometric indicators (weight, height, BMI), estimated glomerular filtration rate (using the CKD-EPI algorithm), performance status, and tumour location. 

As previously reported for genotyping [11], phenotyping in our study allowed us to decrease the dose and the toxicity of 5-FU.

Interestingly, this study is at odds with recent research. For example, the Ontario Health study concluded an uncertain benefit in preventing severe toxicity by dose reduction in DPD deficient patients. This Canadian work was made with genotype-guided fluoropyrimidine dose reduction in heterozygous *DPYD* variant carriers [35], which could explain the different results between our study and this one.

A recent study compared the prevalence of toxicities between patients with and without DPD deficiency, screened by uracil plasmatic determination [36]. It demonstrated no significant difference in the prevalence of toxicities between DPD-deficient and non-deficient patients, suggesting that further work is needed to investigate the association of phenotyping with toxicity [36]. With a different approach, our study demonstrated the contribution of phenotyping in a single laboratory to limit inter-laboratory variability in preventing severe toxicity. Directly comparing toxicities in deficient versus non-deficient patients does not show differences because of the dose adjustments ethically made in deficient patients. Our approach is finally appropriate to demonstrate the effectiveness of a public health strategy by comparing two eras and two practices.

A partial DPD deficiency was found in about 15.7% of the included patients. This observation is higher than the proportion usually described in the general population. As described in the literature, using genotyping, the prevalence of partial deficiency in Caucasians is estimated between 3 and 8%, and that of complete deficiency is between 0.01 and 0.5% [9,12,13,14]. This difference confirms that genotyping is less sensitive than phenotyping. It may also be due to the generalisation of DPD phenotyping after the HAS recommendations with initial difficult compliance to the pre-analytical conditions. It could also come from the recruitment of our patients. The patients included had a history of smoking and alcoholism as well as supportive care medications (proton pump inhibitor), which could impact the functionality of the DPD [37]. Moreover, hepatic disorders frequently observed in these diseases could also have impacted the DPD activity status [38]. In addition to inter-laboratory variability, renal failure, food consumption, circadian rhythm, and pre-analytical errors can have an impact on uracil levels [39]. In the present work, the pre-analytical conditions needed to avoid altering plasma uracil concentrations were fully respected, and samples not respecting these conditions were cancelled and resampled.

In a Dutch prospective study, significant between-centre differences were observed in uracil levels, and no association could be found between pre-treatment uracilemia and DPD activity in peripheral blood mononuclear cells or fluoropyrimidine-related toxicity [39]. To improve the sensitivity and specificity of the test, the accuracy of phenotyping procedures between laboratories should be monitored and standardised [39]. 

Contrarily to the mandatory pre-therapeutic determination of the DPD status, there are no strict guidelines in France for the management of fluoropyrimidines dose reduction in partially deficient patients. That is why this study showed a heterogenicity of medical practices. At the initiation of fluoropyrimidines treatment, no 5-FU dose adjustment was observed for 19% of deficient patients for the first-course bolus and 29% for the infusion. Moreover, for some patients, treatment was not adjusted through the four first courses of treatment. In the second cycle, usually time for upwards dose adjustments when the initial diminished dose was well tolerated, the study described an important abstention. This absence of adjustment could be the result of 5-FU toxicity, but considering our results on acute severe side effects, non-optimal tailoring could be imagined. In deduction, the heterogeneity of the adjustment protocol may subsequently increase the risk of treatment failure.

A recent study [40] suggested that reducing dosage via phenotyping information may reduce toxicity but may also reduce treatment efficacy. Other studies also reported that adjusting dosages according to the level of DPD activity reduces toxicity without decreasing efficacy [2,18,41] or even improves efficacy if combined with pharmacological or therapeutic drug monitoring associated or not with DPD testing [40,41]. Using a retrospective framework, our study was designed to capture severe toxicities. The assessment of clinical response is missing due to the incomplete collection of efficacy data; therefore, our study cannot answer this question. Furthermore, this point would be challenging to evaluate because of a heterogeneous population, particularly regarding the type of tumour, and the indication for chemotherapy (adjuvant or palliative).

To improve the initiation of fluoropyrimidines treatment in DPD deficient patients, pharmacokinetics Bayesian tools could be used [41,42,43]. It is well known that plasma concentrations of 5-FU and capecitabine are associated with both side effects and responses. In addition, studies show that dose adjustment of 5-FU guided by estimation of individual pharmacokinetic parameters can minimise drug-related toxicity and improve the therapeutic index [44,45]. 

Therefore, as proposed by Dolat et al., therapeutic drug monitoring of 5-FU in DPD deficient patients could be conducted to improve practices [40] to avoid the risk of under-exposure and inefficiency of 5-FU. 

This study had several limitations. It was a retrospective, multicentric study but focused on a single French region and the same multidisciplinary board meeting. 

Information about the patients’ *DPYD* genetic background is missing from our study as it is not recommended in France. The lack of genetic data constrains the evaluation of the correspondence between the two predictive markers.

Our study did not evaluate the clinical utility, effectiveness, and cost-effectiveness ratio as it was not designed for this. 

DPD-phenotyping was only based on the measurement of uracilemia, but a recent study [46] described no clear correlation between uracilemia and DPD phenotype, with a risk of toxicity in patients with uracilemia below or near 16 ng/mL, or undertreatment in patients with uracilemia above or near 16 ng/mL. More studies will probably be needed to define the best marker for DPD phenotyping.

At least, it is difficult to compare our work with the numerous international studies based only on genetic analysis as a marker of DPD phenotype.

The outcome of our investigation may be adversely affected by non-homogeneous heterogeneity variables relating to the patients’ characteristics and the administration of their medication (e.g., dose, number of cycles). A prospective randomised controlled study would be needed to confirm our results.

We could not find any difference between the two groups regarding some clinical parameters. For example, the frequency of postponing treatment is only different after three cycles. Toxicity may be cumulative, which requires several cycles in our population to detect it. Regarding treatment discontinuation, we found a tendency for a difference only after the third course of treatment. Dose adjustments between each course are left to the discretion of the individual physician and the lack of power to show a difference partly explains why only toxicities of grades greater than three combined with their frequency are significantly reduced by phenotyping. The retrospective nature of the study also limits the sensitivity to detect toxicities below grade 3.

As expected with the management of fluoropyrimidines in DPD deficient patients, this study showed that the dose intensity of 5U is inferior in deficient than non-deficient patients. However, our study does not allow us to correlate this decrease in dose to a diminution in survival. A large prospective study is ongoing, investigating the effect of phenotype-guided dosing based on pre-treatment uracil level (clinicaltrials.gov identifier NCT04194957).

## 5. Conclusions

This study showed, based on real data, that the new French guidelines limit severe toxicities due to 5-FU during the four first cycles of chemotherapy. For the management of fluoropyrimidines in deficient patients, we observed heterogeneity of medical practices and a significant presence of severe side effects, both of which could increase the risk of therapeutic failure. In this sense, we propose the use of fluoropyrimidines therapeutic drug monitoring in deficient patients.

## Figures and Tables

**Figure 1 pharmaceutics-14-02119-f001:**
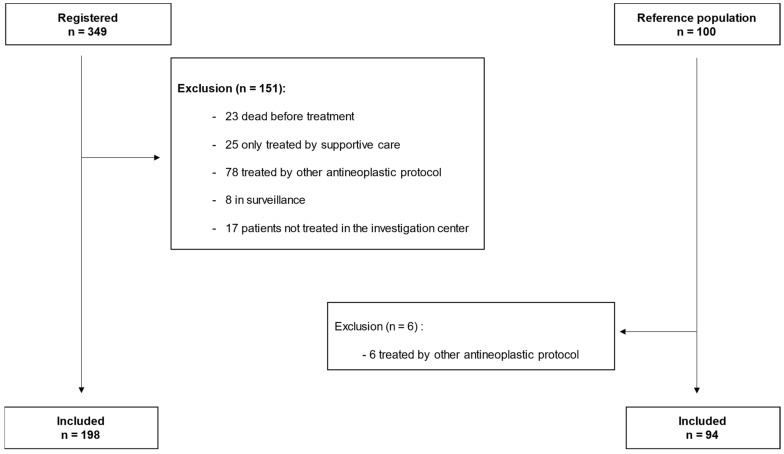
Flow chart of the study. *n*: number.

**Figure 2 pharmaceutics-14-02119-f002:**
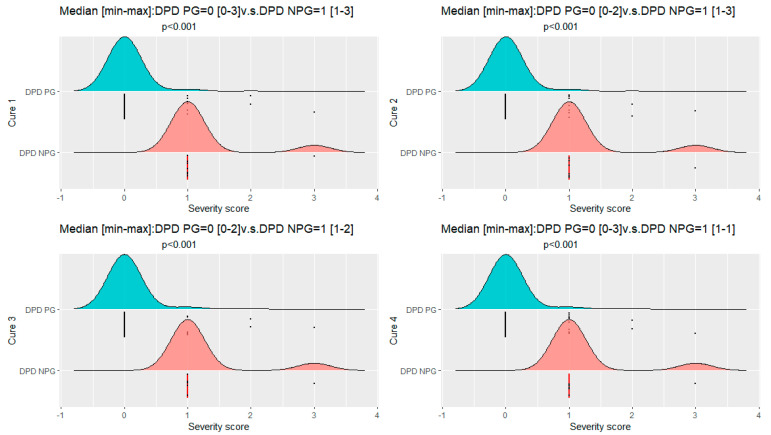
Comparison of severity scores at the different treatment courses between DPD PG and DPD NPG groups. DPD PG: Dihydropyrimidine dehydrogenase phenotype group; DPD NPG: Dihydropyrimidine dehydrogenase no phenotype group; *n*: number.

**Figure 3 pharmaceutics-14-02119-f003:**
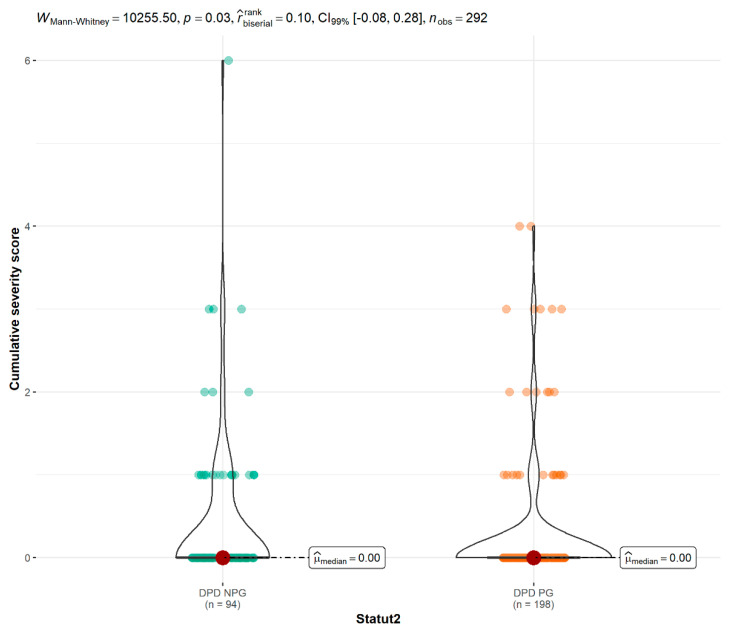
Comparison of cumulative severity scores between DPD PG and DPD NPG groups. DPD PG: Dihydropyrimidine dehydrogenase phenotype group; DPD NPG: Dihydropyrimidine dehydrogenase no phenotype group; *n*: number.

**Table 1 pharmaceutics-14-02119-t001:** Clinical characteristics of the patients.

Characteristic	DPD PG (*n* = 198)	DPD NPG (*n* = 94)	*p* Overall
Age (years) mean [min–max]	66.8 [38.1–88.4]	62.6 [28.7–88.2]	0.004
Gender *n*			0.213
Male	120	49	
Female	78	45	
Stature (cm) [min–max]	170 [142–196]	170 [142–196]	0.986
Weight (kg) [min–max]	70.0 [39.0–137]	69.0 [40.0–151]	0.749
Body surface area (kg/m^2^) [min–max]	24.3 [14.3–43.8]	23.9 [14.8–59.0]	0.618
GFR CKD-EPI (mL/min) [min–max]	90.0 [22.0–144]	97.0 [24.0–141]	0.071
Primary tumoral location *n* (%)			0.080
Colon-rectum	85 (42.9%)	45 (47.9%)	
Pancreas	55 (27.8%)	22 (23.4%)	
Stomach	20 (10.1%)	2 (2.13%)	
Oesophagus	24 (12.1%)	13 (13.8%)	
Neuro-endocrine	9 (4.55%)	9 (9.57%)	
Small intestine	5 (2.53%)	3 (3.19%)	
Treatment type *n* (%)			0.001
Neoadjuvant	50 (25.3%)	23 (24.5%)	
Adjuvant	39 (19.7%)	4 (4.26%)	
Palliative	109 (55.1%)	67 (71.3%)	
Chemotherapy protocol *n* (%)			0.014
FLOT	8 (4.04%)	0 (0.00%)	
FOLFIRI	13 (6.57%)	16 (17.0%)	
FOLFIRINOX	62 (31.3%)	21 (22.3%)	
FOLFOX	90 (45.5%)	41 (43.6%)	
LV5FU2	17 (8.59%)	9 (9.57%)	
LV5FU2 DBZ	8 (4.04%)	7 (7.45%)	
Irinotecan:			0.909
No	123 (62.1%)	57 (60.6%)	
Yes	75 (37.9%)	37 (39.4%)	
Oxaliplatin:			0.009
No	38 (19.2%)	32 (34.0%)	
Yes	160 (80.8%)	62 (66.0%)	
Biotherapy:			0.962
No	160 (80.8%)	75 (79.8%)	
Yes	38 (19.2%)	19 (20.2%)	
Radiotherapy:			0.097
No	168 (84.8%)	87 (92.6%)	
Yes	30 (15.2%)	7 (7.45%)	

DPD P: Dihydropyrimidine dehydrogenase phenotype group; DPD NPG: Dihydropyrimidine dehydrogenase no phenotype group; *n*: number; GFR CKD-EPI: glomerular filtration rate (CKD-EPI algorithm).

**Table 2 pharmaceutics-14-02119-t002:** Treatment postponement, 5-FU pursuit, severity score during the treatment.

Characteristic	DPD PG (*n* = 198)	DPD NPG (*n* = 94)	*p* Overall
Treatment 1: 5-FU dose (bolus + pump)	2800 [400;4000]	2800 [2000;2800]	0.060
Treatment 1: Postponement			0.313
Yes	24 (12.1%)	7 (7.45%)	
No	174 (87.9%)	87 (92.6%)	
Treatment 1: 5-FU discontinuation			0.723
Yes	7 (3.54%)	2 (2.13%)	
No	191 (96.5%)	92 (97.9%)	
Treatment 1: severity score median [min–max]	0.00 [0.00;3.00]	1.00 [1.00;3.00]	<0.001
Treatment 2: 5-FU dose (bolus + pump)	2800 [0.00;4000]	2800 [0.00;2800]	0.257
Treatment 2: Postponement			0.606
Yes	22 (11.5%)	8 (8.70%)	
No	169 (88.5%)	84 (91.3%)	
Treatment 2: 5-FU discontinuation			0.280
Yes	8 (4.19%)	1 (1.09%)	
No	183 (95.8%)	91 (98.9%)	
Treatment 2: severity score median [min–max]	0.00 [0.00;2.00]	1.00 [1.00;3.00]	<0.001
Treatment 3: 5-FU dose (bolus + pump)	2800 [0.00;4400]	2800 [0.00;2800]	0.352
Treatment 3: Postponement			0.012
Yes	19 (10.2%)	1 (1.09%)	
No	167 (89.8%)	91 (98.9%)	
Treatment 3: 5-FU discontinuation			0.067
Yes	12 (6.45%)	1 (1.09%)	
No	174 (93.5%)	91 (98.9%)	
Treatment 3: severity score median [min–max]	0.00 [0.00;2.00]	1.00 [1.00;2.00]	<0.001
Treatment 4: 5 FU dose (bolus + pump)	2800 [0.00;4400]	2800 [0.00;2800]	0.146
Treatment 4: severity score median [min–max]	0.00 [0.00;3.00]	1.00 [1.00;1.00]	<0.001
All Treatment: severity score median [min–max]	0.00 [0.00;4.00]	0.00 [0.00;6.00]	0.028

DPD PG: Dihydropyrimidine dehydrogenase phenotype group; DPD NPG: Dihydropyrimidine dehydrogenase no phenotype group; *n*: number.

**Table 3 pharmaceutics-14-02119-t003:** Acute severe toxicities encountered (NCI grading).

	DPD PG (*n* = 198)				DPD NPG (*n* = 94)			
	Cycle 1	Cycle 2	Cycle 3	Cycle 4	Cycle 1	Cycle 2	Cycle 3	Cycle 4
Acute toxicity events *n*								
Neutropenia G3	3	2	4	1	4	1		1
Neutropenia G4		1	1			1		
Anemia G3	1	3	1	1	1			
Anemia G4			1					
Thrombocytopenia G3	1	2						
Thrombocytopenia G4		1		1				
Nausea G3	1	2		1	1	2	3	
Nausea G4	1				1	1	1	
Diarrhea G3	5	1	1	3	1	4	3	3
Diarrhea G4	1							
Mucite G3				1	1	1	2	

NCI: National Cancer Institute; DPD PG: Dihydropyrimidine dehydrogenase phenotype group; DPD NPG: Dihydropyrimidine dehydrogenase no phenotype group; *n*: number.

**Table 4 pharmaceutics-14-02119-t004:** Dosage adjustment of 5-FU bolus during the first four chemotherapy treatment cycles in deficient patients (*n* = 31).

5-FU Dose	Cycle 1 *n* (%)	Cycle 2 *n* (%)	Cycle 3 *n* (%)	Cycle 4 *n* (%)
Patients *n*	31	29	27	25
Standard dosage	6 (19%)	11 (38%)	14 (52%)	14 (56%)
−25%	16 (52%)	10 (34%)	5 (19%)	3 (12%)
−50%	3 (10%)	1 (3%)	2 (7%)	3 (12%)
−75%	0 (0%)	0 (0%)	0 (0%)	0 (0%)
−100%	6 (19%)	7 (24%)	6 (22%)	5 (20%)

*n*: number.

**Table 5 pharmaceutics-14-02119-t005:** Dosage adjustment of infusional continuous 5-FU during the first four chemotherapy treatment cycles in deficient patients (*n* = 31).

5-FU Dose	Cycle 1 *n* (%)	Cycle 2 *n* (%)	Cycle 3 *n* (%)	Cycle 4 *n* (%)
Patients *n*	31	29	27	25
Standard dosage	9 (29%)	17 (59%)	20 (74%)	20 (80%)
−25%	21 (68%)	12 (41%)	7 (26%)	4 (16%)
−50%	1 (3%)	0 (0%)	0 (0%)	1 (4%)
−75%	0 (0%)	0 (0%)	0 (0%)	0 (0%)
−100%	0 (0%)	0 (0%)	0 (0%)	0 (0%)

*n*: number.

## Data Availability

The data presented in this study are available on request from the corresponding author.

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
