# Peer review of "Impact of Guidelines Regarding Dihydropyrimidine Dehydrogenase (DPD) Deficiency Screening Using Uracil-Based Phenotyping on the Reduction of Severe Side Effect of 5-Fluorouracil-Based Chemotherapy: A Propension Score Analysis"

_pharmaceutics, 2022, doi:10.3390/pharmaceutics14102119_

Round 1
Reviewer 1 Report
In this article, the authors examined the role of DPD deficiency assessment by pretherapeutic testing of plasmatic uracil concentration in preventing fluoropyrimidine-related toxicity. The topic could be of interest and may provide data to support the implementation of uracil testing in the routinely clinical practice. The sample size is adequate. The strengths and limitations of the study are well-discussed. The evaluation of the statistical approach is out of my area of expertise.
Some comments:
- The overall English needs to be extensively edited. Many statements are confusing and hard to understand.
- In lines 69-70 the authors reported three DPYD variants. However, four DPYD variants (DPYD*2A, rs3918290; DPYD*13, rs55886062; c.2846A>T, rs67376798; c.1236G>A-HapB3; rs56038477) are associated with full or partial DPD loss and are currently validated for their clinical impact on fluoropyrimidine-related toxicity. The authors should upgrade the sentence.
- The authors did not consider the DPYD genetic background of the patients (specifically for the four validated markers). It would be interesting to have the genetic profile of patients beyond the uracil plasma levels in order to evaluate the correspondence between the two predictive markers.
- The authors should stress more the need for validating their results in further well-designed studies with homogeneous clinical intervention based on uracil concentration. In fact, the authors recognized that there was a high heterogeneity in the management of fluoropyrimidine dose in partial deficient patients and this could negatively impact the global results of the study.
- Since the high variability between laboratories in the assessment of uracil concentration, the authors should stress more that the analysis is centralized in one laboratory.
- A recent study (in addition to the one mentioned in the text), reported that DPD phenotyping by UH2/U is poorly related to DPD activity measured in peripheral blood mononuclear cells, which is considered the gold standard for DPD phenotyping, and could be strongly influenced by blood collection time and sample processing [M. de With, J. Clin Pharmacol Ther (2022). doi: 10.1002/cpt.2608]. The authors shoul discussed also this paper.
- The authors did not find an association with all the clinical parameters evaluated. This should be better discussed and clinically contextualized.
Author Response
In this article, the authors examined the role of DPD deficiency assessment by pretherapeutic testing of plasmatic uracil concentration in preventing fluoropyrimidine-related toxicity. The topic could be of interest and may provide data to support the implementation of uracil testing in the routinely clinical practice. The sample size is adequate. The strengths and limitations of the study are well-discussed. The evaluation of the statistical approach is out of my area of expertise.
Some comments:
- The overall English needs to be extensively edited. Many statements are confusing and hard to understand.
Author : we thank the referees for their careful review, positive feedback and valuable comments which helped us improve the quality of our manuscript. We thank you for your suggestion. We have checked the entire manuscript by colleague co-authors who are fluent in English. The changes are highlighted in the manuscript. Modifications are highlighted in colour in the manuscript.
In lines 69-70 the authors reported three DPYD variants. However, four DPYD variants (DPYD*2A, rs3918290; DPYD*13, rs55886062; c.2846A>T, rs67376798; c.1236G>A-HapB3; rs56038477) are associated with full or partial DPD loss and are currently validated for their clinical impact on fluoropyrimidine-related toxicity. The authors should upgrade the sentence.pt.
Authors: We thank you for your suggestion. We have listed the most frequently researched. We completely agree with you and are updating the sentence: “Deficiency screening can be performed by genotyping (search for variants DPYD*2A, rs3918290; DPYD*13, rs55886062; c.2846A>T, rs67376798; c.1236G>A-HapB3; rs56038477). ”
- The authors did not consider the DPYD genetic background of the patients (specifically for the four validated markers). It would be interesting to have the genetic profile of patients beyond the uracil plasma levels in order to evaluate the correspondence between the two predictive markers.
Authors: This study was launched following the formalisation by the health authorities of the obligation to phenotype patients before prescribing Fluoropyrimidines. We regret the lack of genetic data, which is a limitation of our study. Only phenotyping is reimbursed or covered by the health insurance scheme. Other studies are in progress to confirm the contribution of the different strategies. We also add these sentences in the limitation of the study:
“Information about the DPYD genetic background of the patients is missing from our study. The lack of genetic data constrains the evaluation of the correspondence between the two predictive markers”.
- The authors should stress more the need for validating their results in further well-designed studies with homogeneous clinical intervention based on uracil concentration. In fact, the authors recognised that there was a high heterogeneity in the management of fluoropyrimidine dose in partial deficient patients and this could negatively impact the global results of the study.
Authors : We agree entirely with your comment. We propose to add the following sentence to the limitations: “The outcome of our investigation may be adversely affected by non-homogeneous heterogeneity variables relating to the characteristics of the patients and the administra-tion of their medication (e.g. dose, number of cycles...). A prospective randomised con-trolled study will allow us to confirm our results.”
Regrettably, the possibilities of conducting a prospective randomised phenotyping and/or genotyping study vs. non-phenotyping or non-genotyping are limited. We presented our data as a case-control study since all French patients are supposed to be tested since 2019 and it is less possible to make a prospective, randomised trial.
Since the high variability between laboratories in the assessment of uracil concentration, the authors should stress more that the analysis is centralized in one laboratory.
Authors: As proposed, we indicate this in the material and method paragraph: “ Uracil measurements were determined according to the latest French guidelines and performed in a single laboratory to limit inter-laboratory variability. ”
Preanalytical conditions for uracil concentration estimation were rigorous. It implied the use of specific material (sample tubes without separating gel and with lithium hep-arinate as an anticoagulant), a limited time between sampling and centrifugation (<2h), transport conditions respecting the cold chain and freezing of the sample (-80°C) immedi-ately after plasma separation.
At the Department of Pharmacology and toxicology, CHU Reims, Plasmatic uracil (U) were quantified using a sensitive ultra-performance liquid chromatography…. .”
- A recent study (in addition to the one mentioned in the text), reported that DPD phenotyping by UH2/U is poorly related to DPD activity measured in peripheral blood mononuclear cells, which is considered the gold standard for DPD phenotyping, and could be strongly influenced by blood collection time and sample processing [M. de With, J. Clin Pharmacol Ther (2022). doi: 10.1002/cpt.2608]. The authors should discussed also this paper.
Author: The interesting article you mention is already cited in reference 40. It has been mentioned in the discussion and we have also updated this part of the discussion as follows:
“….. As with inter-laboratory variability, food consumption and circadian rhythm can have an impact on uracil levels as well as pre-analytical error [40]. In the present work, the pre-analytical conditions limiting the impact of variability factors were fully respected and samples that were taken without respecting these conditions were cancelled and subsequently repeated. ”
In a Dutch prospective study, a large between-center differences was observed in ura-cil levels and the association between pre-treatment uracil and DPD activity in peripheral blood mononuclear cells and fluoropyrimidine-related toxicity could not be found [40]. In order to improve the sensitivity and specificity of the test, the accuracy of phenotyping procedures between laboratories should be monitored and standardised [40].
- The authors did not find an association with all the clinical parameters evaluated. This should be better discussed and clinically contextualized.
Authors : We thank you for your comment. We are aware of this issue and we propose in the limitations paragraph some explanations as follows:
“ We could not find any difference between the two groups regarding some clinical parameters. For example, the frequency of deferral of treatment is different only after three courses of treatment. Toxicity may be cumulative, which requires several treatments in our population to detect it. With regard to discontinuation, we found a tendency to differ only after the third course of treatment. The dose adjustments made between each treatment at the discretion of each physician and the lack of power to show a difference partly explain why only toxicities of grade greater than three combined with their frequency are significantly reduced by phenotyping. The retrospective nature of the study also limits the sensitivity of the protocol to detect toxicities below grade 3. ”

Reviewer 2 Report
This paper by Laures et al. presents Real-World evidence (RWE) data regarding the clinical usefulness (i.e., reducing treatment-related toxicities) of upfront DPD testing in patients scheduled with 5-FU. Their data are presented as a case-control study since all French patients are supposed to be tested since 2019 and it is not possible to make a prospective, randomized trial.
Data are convincing and building a toxicity severity score, the authors demonstrate that indeed, DPD testing significantly reduces the risk of treatment-induced side-effects.
I cannot detect confounding factors or methodological biases in this work, and the propensity score they used is sound - therefore I have only minor issues.
1. Encapsulating the toxicities up to the 4th course is probably too much - usually severe toxicities triggered by gene polymorphisms show at the 1st or the 2d course - at the 4th course, you may have a mix between poor PK due to reduced liver clearance and cumulative toxicities because hematopoietic progenitors start to be exhausted. You are lucky since the difference remains significant, even when considering late-onset toxicities. This should be discussed.
2. Some more details about the methodology used by the prescribers to cut the dosing would be much welcome. Did they use some kind of graphical chart? For instance, the reported U values in the subset of patients with DPD deficiency cover a wide of U>16 ng/ml values, up to 174 ng/ml which is normally associated with precluding any fluoropyrimidine. Still, it seems that only -25% reduction were applied when considering continuous infusion (I am not referring to bolus which is frequently skipped when a patient is even moderately deficient).
3. The primary endpoint was related to the reduction of severe toxicities but comparing the response rate or PFS if available among PG and NPG groups would be interesting. There is a frequent remark with those issues that by reducing the dosing, one may achieve higher safety at the cost of reduced efficacy as well. The authors should discuss this point (that is frequently evoked to justify the absence of recommendations outside France). A couple of papers (Gamelin JCO 1999 and Launay Clin Cancer Drugs 2017) have already suggested that both can be achieved (i.e., reducing the toxs plus maintaining (Launay) or even increasing (Gamelin) efficacy using TDM and/or DPD testing).
Author Response
Review 2
This paper by Laures et al. presents Real-World evidence (RWE) data regarding the clinical usefulness (i.e., reducing treatment-related toxicities) of upfront DPD testing in patients scheduled with 5-FU. Their data are presented as a case-control study since all French patients are supposed to be tested since 2019 and it is not possible to make a prospective, randomized trial.
Data are convincing and building a toxicity severity score, the authors demonstrate that indeed, DPD testing significantly reduces the risk of treatment-induced side-effects.
I cannot detect confounding factors or methodological biases in this work, and the propensity score they used is sound - therefore I have only minor issues.
Authors : We thank the referees for their careful review, positive feedback and valuable comments which helped us improve the quality of our manuscript.
- Encapsulating the toxicities up to the 4th course is probably too much - usually severe toxicities triggered by gene polymorphisms show at the 1st or the 2d course - at the 4th course, you may have a mix between poor PK due to reduced liver clearance and cumulative toxicities because hematopoietic progenitors start to be exhausted. You are lucky since the difference remains significant, even when considering late-onset toxicities. This should be discussed.
Authors: We fully agree that, as reported in several studies, the toxicities preventable by phenotyping are those of the first and second cycle. In our case we identified a difference in toxicity score up to the fourth treatment cycle probably because of the accumulation of drug toxicities, the decrease of physiological capacity of cell regeneration related to the depletion of stem cells, haematopoietic progenitors, and the depletion of cellular detoxification systems against oxidative stress.
We updated the discussion as follow: “Our study shows a significant difference in toxicities during the four cycles. As reported by several studies, the toxicities avoidable by phenotyping are those of the first and second cycles [15,25,36]. In our study, we identified a difference in toxicity score up to the fourth treatment cycle probably due to the accumulation of drug toxicities, the decrease of physiological capacity of cell regeneration related to the depletion of stem cells, haematopoietic progenitors, and the depletion of cellular detoxification systems against oxidative stress. No life-threatening events were observed in either group.”
However, the decrease in clearance of 5-FU is not often confirmed as shown by Gamelin in 1999.
- Some more details about the methodology used by the prescribers to cut the dosing would be much welcome. Did they use some kind of graphical chart? For instance, the reported U values in the subset of patients with DPD deficiency cover a wide of U>16 ng/ml values, up to 174 ng/ml which is normally associated with precluding any fluoropyrimidine. Still, it seems that only -25% reduction were applied when considering continuous infusion (I am not referring to bolus which is frequently skipped when a patient is even moderately deficient).
Authors: Thank you for your comment. In our study, there is no complete deficiency for treeted patients. Uracil concentration median was 19 ng/mL and ranged between 16.1– 52.2 ng/ml. we updated the materials and methods section as follows:
“The French recommendations advocate 5-FU doses tailoring according to the extent of the detected DPD impairment and adjusted based on age, general condition, and other clini-cal/paraclinical covariates if required. In our study, patients were adjusted as follows: full dose when plasma uracil <16 µg/mL; 25% dose reduction when 16 µg/mL<plasma uracil<50 µg/mL; 50% reduction when 50 µg/mL<plasma uracil<100 µg/mL; 75% reduction when 100 µg/mL<plasma uracil<150 µg/mL.”
- The primary endpoint was related to the reduction of severe toxicities but comparing the response rate or PFS if available among PG and NPG groups would be interesting. There is a frequent remark with those issues that by reducing the dosing, one may achieve higher safety at the cost of reduced efficacy as well. The authors should discuss this point (that is frequently evoked to justify the absence of recommendations outside France). A couple of papers (Gamelin JCO 1999 and Launay Clin Cancer Drugs 2017) have already suggested that both can be achieved (i.e., reducing the toxs plus maintaining (Launay) or even increasing (Gamelin) efficacy using TDM and/or DPD testing).
Authors: We are aware of this discussion, and we share the same opinion as yours. We thank you very much for your critical, rich, and very intelligent reading. We completed our discussion as you suggested as follows:
“
A recent study [42] suggested that reducing dosage via phenotyping information may reduce toxicity but may also reduce treatment efficacy. Other studies also reported that adjusting dosages according to the level of DPD activity reduces toxicity without de-creasing efficacy [2,25,43] or even improves efficacy if combined with pharmacological or therapeutic drug monitoring associated or not with DPD testing [42,43]. Using a retrospective framework, our study was designed to capture severe toxicities. The assessment of clinical response is missing due to the incomplete collection of efficacy data and there-fore our study cannot answer this question. This point would be difficult to evaluate be-cause of a heterogeneous population, particularly regarding the type of tumour, the indi-cation for chemotherapy (adjuvant or palliative).”
A large prospective study is ongoing investigating the effect of phenotype-guided dosing based on pretreatment uracil level (clinicaltrials.gov identifier NCT04194957).
